# Optimal First-Line Medico-Surgical Strategy in Ovarian Cancers: Are We There Yet?

**DOI:** 10.3390/cancers15143556

**Published:** 2023-07-10

**Authors:** Stanislas Quesada, Quentin Dominique Thomas, Pierre-Emmanuel Colombo, Frederic Fiteni

**Affiliations:** 1Institut Régional du Cancer de Montpellier (ICM), 34298 Montpellier, France; stanislas.quesada@icm.unicancer.fr (S.Q.); quentin.thomas@icm.unicancer.fr (Q.D.T.); pierre-emmanuel.colombo@icm.unicancer.fr (P.-E.C.); 2Medical Oncology Department, University Hospital of Nîmes, 30900 Nîmes, France

**Keywords:** epithelial ovarian cancer, primary cytoreductive surgery, interval cytoreductive surgery, prognosis, medicosurgical strategy

## Abstract

**Simple Summary:**

Cytoreductive surgery represents the cornerstone of the management of advanced epithelial ovarian cancers (aEOC). Currently, two alternative strategies coexist: either primary cytoreductive surgery (PCS) followed by adjuvant systemic therapy or neoadjuvant chemotherapy (NACT) followed by interval cytoreductive surgery (ICS). Although PCS and ICS exhibit distinct pros and cons, a univocal optimal strategy has not been reached yet. In this state-of-the-art review, we will describe achievements, ongoing data, and perspectives regarding the optimal medicosurgical strategy for aEOC.

**Abstract:**

In spite of tremendous advances in advanced ovarian cancer management through the past decade, notably owing to surgical expertise and novel combination molecules (including bevacizumab and PARP inhibitors), the optimal initial sequential strategy remains a major concern. Indeed, following seminal clinical trials, primary cytoreductive surgery (PCS) followed by adjuvant systemic therapy and interval cytoreductive surgery (ICS) following neoadjuvant chemotherapy (NACT) have been positioned as validated alternatives with distinct pros and cons, although a definite response is still unassessed. In clinical practice, decisions between PCS and ICS rely on multilayer parameters: the tumor itself, the patient, and the health structure. In this state-of-the-art review, we will discuss the current evidence based on clinical trials and real-world data and highlight the remaining questions, including the fittest positioning of PCS vs. ICS and the optimal number of NACT cycles; subsequently, we will discuss current axes of research such as dedicated clinical trials and more global perspectives. These ongoing strategies and perspectives could contribute to improving the patient journey through personalized medicine.

## 1. Introduction

With over 300,000 new cases reported in 2020, ovarian cancers (which include primary tumors of the ovary, fallopian tubes, and peritoneum) are ranked as the 8th most common cancer in women worldwide [1]. In addition, they are the most lethal gynecologic malignancies in women, accounting for over 200,000 deaths. Approximately 70% of patients with epithelial ovarian cancer (EOC) are diagnosed at an advanced stage (aEOC), corresponding to Stage III-IV disease according to the International Federation of Gynecology and Obstetrics (FIGO) classification [2]. In spite of optimal treatment with a multimodal strategy including debulking surgery and platinum-based chemotherapy, the five-year overall survival (OS) rate for these advanced stages remains at approximately 40% [3]. Breakthrough progress has been achieved regarding maintenance treatments, including bevacizumab and poly[ADP-ribose] polymerase inhibitors (PARPi) [4,5,6,7,8]. To date, the optimal strategy regarding the sequential approach between surgery and chemotherapy remains controversial. Indeed, primary cytoreductive surgery (PCS) followed by adjuvant chemotherapy (ACT) was considered the gold standard for a long time. Nevertheless, several randomized controlled trials have challenged this dogma, positioning neoadjuvant chemotherapy (NACT) and interval cytoreductive surgery (ICS) as relevant alternatives [9,10,11]. To date, a definite proof of the optimal strategy has not been achieved, and decisions should be made according to distinct parameters that will be dissected in this manuscript.

In this state-of-the-art review, after developing rationale and current evidence regarding the choice between these two strategies, we will discuss currently unassessed questions, including the gray zone between PCS and ICS, the optimal number of NACT cycles, the inclusion of complementary treatments within the global strategy, molecular features, and the patient’s perspective.

## 2. Compelling Evidence and Distinct Rationales between Primary and Interval Cytoreductive Surgeries

### 2.1. Pros and Cons of Distinct Strategies

Cytoreductive surgery represents a cornerstone of multimodal treatment of aEOC, nevertheless requiring no residual disease to achieve a maximal impact on the OS of patients [12]. Since its first description in 1975, the definition of optimal cytoreductive surgery has evolved and is currently defined as no macroscopically visible disease (CC0) or <2.5 mm residues (CC1); conversely, CC2 and CC3 represent 2.5–25 mm and >25 mm residual tumor, respectively [13].

Numerous studies have demonstrated the major prognostic role of optimal cytoreductive surgery in OS. A seminal meta-analysis reported that each 10% increase in the rate of optimal PCS was associated with a 5.5% improvement in OS [14]. Subsequently, it was shown that the best results were observed after complete resection of abdominopelvic lesions without macroscopic tumor residue at the end of surgery. This complete surgery has become the mainstay for aEOC, a requirement highlighted in international guidelines [2,9,10,11]. PCS exhibits notable advantages. By removing all visible disease, it favors the diffusion of ACT into microscopic residual disease [15]. Indeed, neoplastic neoangiogenesis induces hypoxic zones and intratumoral ischemia, which are more important in supra-centimeter lesions. This neoangiogenesis reduces the intracellular penetration of systemic treatments, particularly in the peritoneum. Furthermore, drastically decreasing tumor volume before chemotherapy challenges reduces the probability of the emergence of resistant clones and thus reduces the risk of tumor recurrence [16]. The main concern regarding PCS is its potent morbidity, as it may include digestive resections, partial peritoneum removal, and lymphadenectomy. Validated scores exist to assess tumor extension and help optimize the therapeutic strategy between PCS/ACT and NACT/IDS (Table 1). Indeed, while FIGO staging relying on thoraco-abdomino-pelvic tomodensitometry is of prime importance for prognosis and strategic perspectives, it fails to accurately predict the extension of carcinomatosis [17]. Apart from evident localized metastatic (e.g., lung/liver) diseases, initial laparoscopy remains the gold standard for initial exploration of the peritoneal cavity [9,10,11,18]. Apart from technically feasible CC0 surgery, quality of life (QoL) is currently of main concern when considering current progression-free survival (PFS) and OS in an aEOC. On the other hand, NACT plus ICS relies on decreasing initial tumor burden, allowing distinct advantages: technically easier cytoreductive surgery, fewer sequelae, and the potential to render patients operable. Notably, ICS may be a reasonable alternative for patients with stage IV disease and/or poor performance status. Furthermore, ICS may lead to improved QoL in specific populations [19].

As for many “two strategies” dilemmas, several studies and meta-analyses have been published so far, emphasizing distinct advantages [20]. Indeed, orientation between PCS and ICS relies on randomized clinical trials (RCT) and real-world data (RWD), which bring distinct information.

**Table 1 cancers-15-03556-t001:** Principal validated surgical scores in the management of aEOC.

	Fagotti Score [21]	Makar Classification [22]		Peritoneal Cancer Index ^1^ [23]	
Parameters	Score	Class	Disease Localization	Region	Localization
Infiltration of greater omentum	2: diffuse stomach infiltration 0: isolated sites	1	Disease located to pelvis Few/no ascites No need for bowel resection	1	Central
				1	Right upper
Peritoneal carcinomatosis	2: non-resectable carcinomatosis or miliary 0: limited-area carcinomatosis (e.g., gutter, resectable by peritonectomy)	2	Disease located to pelvis Few/no ascites One digestive resection necessary	2	Epigastrium
				3	Left upper
Diaphragmatic carcinomatosis	2: widespread infiltration or confluent nodules 0: all other cases	3	Mainly supra-mesocolic Few/no ascites No need for bowel resection	4	Left flank
				5	Left lower
Mesenteric retraction	2: yes 0: no	4	Mainly supra-mesocolic Few/no ascites One digestive resection necessary	6	Pelvis
				7	Right lower
Bowel infiltration	2: gastrointestinal resection is envisioned 0: all other cases	5	Mainly supra-mesocolic Abundant ascites/miliary Several digestive resections necessary	8	Right flank
				9	Upper jejunum
Stomach infiltration	2: nodules infiltrating the stomach and/or spleen and/or lesser omentum 0: all other cases			10	Lower jejunum
				11	Upper ileum
Hepatic metastases	2: any tumor with an area > 2 cm 0: all other cases			12	Lower ileum


^1^ Peritoneal Cancer Index (PCI), also known as “Sugarbaker’s PCI”, relies on combining cancer implant size with their distribution throughout 13 abdominopelvic regions to produce a quantitative score. Each of the 13 regions (0–12) receives a score from 0 to 3, depending on the largest implant: 0, no implants; 1, implants < 0.5 cm; 2, 0.5 to 5 cm; 3, ≥5 cm (or confluent disease or tissue adhesions).

### 2.2. RCT’s Data

Four multicenter, phase III RCTs that directly compared PCS vs. ICS have been published so far (Table 2). Three of these trials were designed as “non-inferiority” (EORTC 55971, CHORUS, and JCOG) as a way to validate ICS as an equivalent alternative to PCS, while the last one (SCORPION) used the “superiority” hypothesis.

The European Organization for Research and Treatment of Cancer (EORTC) 55971 trial (NCT00003636) and the Medical Research Council Chemotherapy Or Upfront Surgery (CHORUS) trial (ISRCTN74802813) included patients with FIGO stages III/IV and compared PCS plus six cycles of ACT (which consisted of platinum salt doublet or platinum salt monotherapy) versus three cycles of NACT plus ICS plus three cycles of ACT [24,25]. These two trials concluded that the NACT strategy was non-inferior to PCS. Indeed, OS was 30.0 and 29.0 months, respectively (EORTC 55971; hazard ratio [HR] 0.98 [90% confidence interval (CI) 0.84–1.13]) and 24.1 and 22.6 months (CHORUS trial; HR 0.87 [95% CI 0.72–1.05]). In multivariate analyses, completeness of surgery appeared to be the strongest independent prognostic factor associated with improvement in OS, both for PCS and ICS strategies. Thus, these two trials reinforced the hypothesis that NACT may participate in achieving complete surgery. Indeed, these trials reported an increase in the rate of complete surgery with the use of a NACT, from 19.4% and 17% to 51.2% and 39% (EORTC and CHORUS, respectively). Furthermore, postoperative morbidity and mortality were lower when NACT was performed; following the same trend, a significant reduction in the rate of multiple bowel resections allowed by NACT contributed to a decrease in the rate of post-operative complications. Nevertheless, both the EORTC and CHORUS trials have been criticized for the low quality of surgical management, with a very low rate of complete surgery and poor OS when compared to other reports and RWD. For instance, in the CHORUS trial, bilateral oophorectomy and hysterectomy were realized in 73% and 76%, respectively. Furthermore, less than 20% of patients received upper abdominal surgery, resulting in a median time for surgery of around 120 min. As such, although cytoreductive surgery for aEOC is considered a procedure requiring surgical and structural expertise, these trials appeared to target “less-experienced surgeons”. Subsequently, a pooled analysis of these two trials with stratification by FIGO stage was performed [19]. It appeared that patients with FIGO stage IV had significantly better outcomes with a NACT/ICS strategy (versus PCS, respectively): median OS was 24.3 versus 21.2 months (HR 0.76 [95% CI 0.58–1.00]; *p* = 0.048) and median PFS was 10.6 versus 9.7 months (HR 0.77 [95% CI 0.59–1.00]; *p* = 0.049). Conversely, when focusing on FIGO IIIc patients with the largest lesions <5 cm, PFS was lower with NACT/ICS versus PCS (11.7 versus 12.2 months; HR 1.26 [95% CI 1.06–1.75]; *p* = 0.017); nevertheless, this difference had no impact on OS (30.2 versus 33.0 months, respectively; HR 1.26 [95% CI 0.96–1.65]; *p* = 0.092).

The third non-inferiority trial, using a Japanese patient cohort (JCOG), was also conducted with overall survival as the primary endpoint [26]. The study design was similar to the previous two trials, however, with a total of eight cycles of perioperative chemotherapy. Median overall survival was 49.0 months and 44.3 months in the PCS vs. ICS, respectively (HR 1.052 [90.8% CI 0.835–1.326]; *p* = 0.24). The lack of demonstration of the non-inferiority of NACT in this trial outweighed the results of the EORTC/CHORUS trials.

The main weakness of these three studies resides in the low rate (i.e., less than 20%) of primary surgery with zero macroscopic residue (CC0) or with tumor residue of less than one centimeter (=optimal resection) [28]. Two factors may have contributed to this non-optimal result: the absence of systematic preoperative evaluation of tumor extent with surgical inspection and the possibility of performing cytoreductive surgery in non-expert centers. For instance, when performed in expert centers, the rate of optimal resection surgery is around 50–70% [29]. As such, the low rates of optimal resection rates in the EORTC (19.4%) and the CHORUS (17%) trials may have explained the poorer OS in PCS arms (29 months and 22.6 months, respectively) compared with data from other previously published studies. For example, in a combined analysis of the AGO OVAR 3, 5, and 7 trials (published during the 2003–2006 period), patients with PCS exhibited a mean OS of 44.1 months [95% CI 42.3–46.4], accompanied by an optimal resection rate of 33.5% [30]. Furthermore, patients in the EORTC trial exhibited a worse prognosis than patients in many other series, even those with optimal resection. Other factors may have been loose inclusion criteria, such as the absence of an age limit (the median age was 62 years [25–86 years]) and the possibility of including patients with a performance status (PS) of 2 (11.9% of patients included); performing PCS (implying a large cytoreduction process) in this kind of already-fragile patient may have negatively impacted OS.

In 2020, the SCORPION (NCT01461850) trial was published, attempting to provide some answers to this controversy [27]. This phase III, randomized, monocentric, superiority trial was conducted in an Italian expert center (Sacred Heart Clinical Trial Center, Roma) and included 147 patients with FIGO stages IIIc/IV [18]. The rationale for developing a monocentric RCT was to circumvent the low accrual rate per center exhibited in previously discussed trials. The primary endpoint was the median PFS between NACT/IDS and PCS/ACT. Strikingly, compared to the EORTG and CHORUS trials, the optimal resection rate reported here was 47.6% versus 77.0% (NACT and PCS arms, respectively). There was no difference regarding PFS, with 14 versus 15 months, respectively, in the NACT and PCS arms (HR 1.05 [95% CI 0.77–1.44]; *p* = 0.73). Furthermore, when focusing on patients without macroscopic residue, no statistically significant differences were observed, neither for PFS (HR 1.01 [95% CI 0.66–1.55]; *p* = 0.96) nor for OS (HR 1.10 [95% CI 0.61–1.97]; *p* = 0.74). Noteworthy, NACT led to a significant reduction in early (<1 month) and late (up to six months) postoperative toxicities. It should be noted that the same Italian team conducted a phase II multicentric study, the MISSION trial (NCT 02324595), between December 2013 and February 2015 [31]. The objective was to evaluate the feasibility and early complication rate of minimally invasive ICS for aEOC patients with clinical complete response after NACT. From 184 patients considered eligible for ICS, 52 (28.2%) met inclusion criteria, and 30 (16.3%) received the planned minimally invasive ICS. The median operative time was 285 min (range 124–418); the median estimated blood loss was 100 mL (range 50–200). CC0 surgery was reached for 29 (96.6%) patients. No early postoperative complications were observed. With a median follow-up of 10.5 months, 23 (77.7%) patients had not relapsed. Although the median operative time does not seem to be decreasing with minimally invasive ICS, the early postoperative complications rate seems to be reduced without impacting survival data. The possibility of minimally invasive surgery therefore seems to have a place for hyperselected patients treated with ICS but needs to be confirmed with a larger prospective study with longer follow-up.

### 2.3. Real-World Data (RWD)

Apart from RCTs, RWDs should not be overlooked, notably because of the lack of generalizability of the former in real-life clinical practice. Rather than confronting these two types of information, they should be coupled, as they shed distinct light on better management of patients with aEOC.

Since 2010 (corresponding to the publication of the EORTC trial), a paradigm shift in clinical practice has been observed [24]. On a large cohort of 19,562 patients, including women with a diagnosis of stage IIIc and IV EOC between January 2004 and December 2015, who were followed up through the end of 2018 in the US, a clear increase in the use of NACT from 21.7% in 2004–2009 to 42.2% in 2010–2015 has been observed. Regarding survival outcomes and the postoperative mortality risk, NACT was associated with a reduction in the risk of 30-day mortality without a decrease in median OS, supporting the results from previously described RCTs [32]. Some observational studies that compared outcomes between patients who received PCS versus ICS have concluded that NACT seemed to be associated with worse survival outcomes [33,34,35]. These results are supported by a meta-analysis examining data from 16 trials with 57,450 participants (NACT, 9475; PCS, 47,975). The results showed that NACT achieved more complete debulking removal (HR 1.69 [95% CI 1.32–2.17]; heterogeneity: *p* < 0.001; I^2^ = 81.9%); reduced the risk of post-operative death (HR 0.18 [95% CI 0.06–0.51]; heterogeneity: *p* = 0.698; I^2^ = 0%); but showed significantly reduced OS compared with PCS (HR 1.30 [95% CI 1.13–1.49]; heterogeneity: *p* < 0.001; I^2^ = 82.7%) [36]. Nevertheless, these studies failed to consider the effect of patient-level confounders such as disease burden, functional status, and the impact of tumor biology on the response to treatment. Some studies that evaluated the potential effect of these unmeasured confounders found that the apparent benefit of PCS could be explained by factors such as limited age at diagnosis, better PS, preoperative disease burden, and *BRCA1/2* mutational status [37,38]. For example, a Chinese monocentric retrospective study found in a cohort of 273 patients that lower BMI, normal CA-125 levels before ICS, CC0 cytoreduction, and a shorter interval between preoperative and postoperative chemotherapy were independent factors associated with better PFS for patients treated by ICS [39]. Moreover, a retrospective study of 1,268 patients with aEOC treated from 2010 to 2016 reported that NACT use significantly increased (*p* < 0.05) in older women (i.e., ≥65 years; 48.4% relative increase), followed by FIGO stage IV disease (35.2% relative increase) [40]. Interestingly, both strategies led to similar OS in older women (HR 1.07 [95% CI 0.95–1.20]; *p* = 0.284) and in FIGO stage IV diseases (HR 0.96 [95% CI 0.84–1.10]; *p* = 0.564).

Regarding post-surgical complications, interesting data have been reported in RWD, confirming what was known owing to RCTs. Indeed, NACT compared to PCS seems to exhibit lower rates of overall post-operative serious adverse events [41,42]. A recent meta-analysis (based on 17 studies, including 3759 patients) evaluated the morbidity and mortality in NACT vs. PCS and found that patients in the PCS group were significantly more likely to have higher morbidity (defined as a Clavien-Dindo grade ≥ 3), with an overall rate of 21.2% compared to 8.8% for the ICS group [95% CI 1.9–4.0; *p* < 0.0001] and were more likely to die within 30 days post-surgery (HR 6.1 [95% CI 02.1–17.6]; *p* = 0.0008). Patients who underwent ICS had significantly shorter procedural times (median duration of minus 35 min; *p* = 0.01), lost less blood intraoperatively (median of minus 382 mL; *p* < 0.001), and had an average shorter admission duration (median duration of minus 5 days; *p* = 0.002) than those undergoing PCS [43].

Globally, the main limitation of RWDs is the selection criteria, which may lead to selection bias. Indeed, patients with poor performance status, other medical conditions, frailty, and/or FIGO stage IV disease will preferably be treated with NACT followed by ICS. These clinicopathological characteristics associated with poor prognosis certainly bias the data and may result in an underestimation of the survival outcomes for patients treated with ICS. Some resectability scores have been created to predict CC0 surgery according to clinical and radiologic assessment and help the physician choose between PCS/ACT and NACT/IDS. The Memorial Sloan Kettering Cancer Center team had developed a multimodality triage algorithm. From 406 patients treated between April 2015 and August 2018, 299 patients (74%) met all eligibility criteria. Based on the prospectively calculated resectability score, 226 patients (76%) were designated as low-risk for suboptimal cytoreduction and 73 patients (24%) as high-risk. In the low-risk group, 181 patients (80%) underwent PCS via laparotomy, 43 patients (19%) underwent laparoscopy, and 2 patients (1%) received NACT. For low-risk patients who underwent laparoscopic evaluation, 72% (*n* = 31/43) had PCS, and 28% (*n* = 12/43) were deemed unresectable and triaged to NACT. Of the 212 low-risk patients undergoing PCS, cytoreductive outcomes were as follows: CC0, 168 patients (79%); optimal cytoreduction (≤10 mm residual disease), 199 patients (94%); suboptimal debulking, 13 patients (6%). Of 37 high-risk patients who underwent PCS, cytoreductive outcomes were as follows: CC0, 20 patients (54%); optimal cytoreduction, 35 patients (95%); suboptimal debulking, 2 patients (5%). The multimodal algorithm was followed in 92% of cases (*n* = 275/299). This type of algorithm can help maximize the availability of PCS to the most patients while minimizing risk and futile intervention for aEOC [44].

### 2.4. Apart from Primary versus Interval Surgeries: How Many NACT Cycles?

When NACT is retained, the optimal timing of ICS is still debated. The purpose of NACT is to achieve a significant tumor response and allow for CC0 ICS. International guidelines recommend three NACT cycles, which seems to be a good compromise between a minimal number to obtain a tumor response and surgery as early as possible [9,10,11]. When applied in practice, a fourth NACT cycle is often performed to organize the ICS logistically. In connection with obviously unattainable CC0 ICS after 3–4 NACT cycles, a second possibility appears: either prolong NACT (≥5 cycles) to optimize tumor response and achieve delayed ICS or consider exclusive medical treatment.

Numerous retrospective studies have analyzed the impact of the number of NACT cycles on patients’ outcomes, with divergent results. On the one hand, a greater number of NACT cycles appeared to be associated with a decrease in PFS and OS, despite a noninferior rate of CC0 surgery [45,46,47,48,49,50]. On the other hand, other series argued that delayed ICS did not worsen the prognosis [51,52,53,54,55]. As for PCS and classical ICS, the main objective of delayed ICS is to achieve CC0 surgery, as the presence of tumor residue in this situation appears to severely worsen the patient’s prognosis [52,53]. Regarding toxicities, postoperative morbidity and mortality are analogous between classical and delayed ICS [51,54]. The optimal timing of ICS should also be evaluated based on the expected complexity of the surgical procedure. Some teams define extensive cytoreduction surgery (ECC) as the performance of at least six organ resections from the following eight categories: salpingo-oophorectomy, total hysterectomy, omentectomy, appendectomy, pelvic lymph node resection, aortic lymph node resection, colonic resection, and other organ resection (e.g., spleen, liver, small bowel). Performing ECC does not seem to provide any benefit to OS in univariate analysis (*p* = 0.49) compared to “conventional” standard surgery [56]. Indeed, increasing evidence suggests that ECC is the main cause of postoperative complications and impaired quality of life in patients, with the major risk of morbidity–mortality caused by multiple intestinal resections [57]. Aggressive surgery may also negatively impact survival due to the risk of delay in the initiation of adjuvant treatments [47,58]. The complexity of the timing of ICS relies on identifying the prognostic factors to guide the physician towards the optimal therapeutic strategy. Different prognostic factors have been highlighted in these retrospective studies, such as the preoperative general condition of patients, FIGO staging, histological subtype, CA-125 decay kinetics, number of NACT cycles, sensitivity to NACT, CC0 surgery, absence of gastro-intestinal resection, and postoperative complications [9,10,11].

As a way to assess the optimal timing of ICS, we recently performed a retrospective study attempting to limit the bias of confounding factors influencing the selection of patients for a standard ICS (i.e., 3–4 NACT cycles) or delayed ICS (i.e., ≥5 NACT cycles) using two propensity score statistical methods (the inverse probability of treatment weights, or IPTW method, and the propensity score matching method) on a large French national cohort (the epidemiological strategy and medical economics, or ESME) of 928 aEOC patients [59]. The ESME research program is based on data collected from expert French Comprehensive Cancer Centers, with the aim of providing independent and high-quality RWD collected from electronic health records. The purpose of a propensity score was to mimic the effect of randomization on the selected variables of interest in a retrospective cohort [60]. After a median follow-up of 49.0 months (standard and delayed ICS, respectively), from the IPTW analysis, median PFS were 17.6 and 11.5 months (HR 1.42 [95% CI 1.22–1.67]; *p* < 0.0001), while median OS were 51.2 and 44.3 months (HR 1.29 [95% CI 1.06–1.56]; *p* = 0.0095). From the matched-pair analysis (352 patients for each group), standard ICS was associated with better PFS (HR 0.77 [95% CI 0.65–0.90]; *p* = 0.018) but not significantly associated with better OS (HR 0.84 [95% CI 0.68–1.03]; *p* = 0.0947). As such, we reported interesting clues regarding the negative prognostic impact of performing ICS beyond four NACT cycles.

An important consideration is that, despite rigorous data collection, all RWDs are retrospective and inevitably contain bias in patient selection. Nevertheless, RWD suggests that patients with poor prognostic factors such as impaired general condition, stage IV disease upon diagnosis, or limited chemosensitivity are more likely to be offered a delayed ICS [61]. Thus, it seems difficult to draw firm conclusions regarding the prognostic impact of delayed ICS. Survival data may also vary according to the inclusion periods, taking into account advances in surgical techniques as well as changes in perioperative and maintenance systemic treatments such as bevacizumab and PARPi.

## 3. Ongoing Projects and Perspectives

### 3.1. PCS vs. ICS

Based on the data from RCTs and despite their limitations, international guidelines have positioned PCS as the preferred option for FIGO stages IIIc/IV, high-grade aEOC, unless complete surgery is not considered feasible right from the start [9,10,11]. The notion of “not considered easily feasible encompasses distinct situations, ranging from surgical considerations to patient status. Surgically, accepted criteria for non-resectability are major confluent involvement of the peritoneum or diaphragmatic cupolas, involvement of the hepatic pedicle or intraparenchymal, the impossibility of restoring continuity intraoperatively, total colectomy, the necessity to perform more than two digestive anastomoses, and planned resections that would leave less than 1.5 to 2 m of small bowel [62]. Regarding disease extent, the 5-class Makar classification and Fagotti score are useful parameters to consider in defining the choice between PCS and NACT. These assess the extension of the abdominopelvic tumor volume in cases of aEOC with peritoneal involvement [21,22,63]. Regarding patients, impaired general conditions, including undernutrition, make primary surgery more morbid [9,10,11]. In the context of the presence of at least one of these criteria, NACT should be preferred. Indeed, when tumor extension leads to unfeasible surgical resection without macroscopic residue (CC0), incomplete PCS should not be performed. This assumption is based in particular on the fact that for patients with suboptimal surgery realized in a non-expert center, OS is clearly impaired compared to patients with similar clinicobiological characteristics and with PCS without macroscopic tumor residue. This point sustains the “no surgery is better than an incomplete one” dogma for aEOC [61]. Previously published trials have given specific insights regarding NACT and PCS strategies, but their limitations, such as the recruitment of patients, were not restricted to expert centers. As a way to circumvent this limitation, the ongoing open, randomized, controlled, multi-center phase III TRUST (NCT02828618) trial will assess OS with NACT/ICS versus PCS in FIGO IIIb-IVb considered resectable aEOC patients and treated in highly qualified ovarian cancer surgery centers [64]. The latter is ensured by the fact that participating centers had to meet certain quality criteria, such as a complete resection rate of more than 50% in PCS for FIGO IIIb-IVb aEOC and the performance of at least 36 cytoreductive surgeries per year. Given the current momentum and the need to treat patients with aEOC in expert centers with standardized and robust procedures, the upcoming results of the TRUST trial will help facilitate the choice between PCS and ICS.

### 3.2. Conventionnal vs. Delayed ICS

Regarding the best strategy between conventional and delayed ICS, several points should be considered. Firstly, novel tools could help in the implementation of the surgical strategy. The CA-125 ELIMi-nation Rate ConstantK (KELIM) is a mathematical model based on the analysis of the kinetics of CA-125 decay during chemotherapy [65]. KELIM has been shown to have both predictive and prognostic roles in reducing the risk of platinum-resistant relapse in patients treated in the first line with PCS followed by ACT [66]. The value of KELIM has been highlighted to assess the chemosensitivity of patients receiving NACT in the first treatment line. Patients with a favorable KELIM score (defined as a value >1.00) had a significantly greater probability of CC0 ICS than patients with an unfavorable (<0.50) or intermediate (0.50–1.00) KELIM (area under curve = 0.76 [95% CI 0.68–0.84]). A favorable KELIM score also had significant prognostic value compared with an unfavorable KELIM in OS (HR 0.28 [95% CI 0.16–0.50]); *p* < 0.001 [67]. In clinical practice, KELIM could provide additional assistance in guiding the timing of ICS. Currently, the recommendations of the National Comprehensive Cancer Network (NCCN) suggest the continuation of NACT for at least six cycles in cases of stable unresectable disease after three cycles [10]. Patients who remain unresectable after 3–4 NACT cycles and who have implicitly poor prognostic factors will mostly benefit from delayed ICS. A central question regarding the optimal timing of ICS is whether delaying ICS after six NACT cycles has a prognostic impact in patients who are resectable after 3–4 NACT cycles. In order to propose a more consensual strategy, an RCT and/or a meta-analysis of all this retrospective data seem necessary. There are currently several ongoing clinical trials looking at the optimal number of NACT cycles to be administered for aEOC. Of note, the CHRONO trial (NCT03579394) compares PFS when ICS is performed after three versus six cycles in stage IIIb-IV patients who are inoperable upon diagnosis but considered eligible for ICS after three cycles (i.e., randomization is performed after these three cycles). The GOGER-01 trial (NCT02125513) focuses on the rate of CC0 ICS when performed after three versus six cycles in stage IIIb-IV aEOC patients. Noteworthy, using standard rather than delayed ICS allows earlier maintenance therapy introduction (i.e., bevacizumab +/− PARPi).

### 3.3. Hyperthermic Intraperitoneal Chemotherapy

Owing to the fact that disease recurrence primarily involves the peritoneal cavity, hyperthermic intraperitoneal chemotherapy (HIPEC), complementary to cytoreductive surgery, has been developed. HIPEC relies on enhanced delivery of chemotherapy at the peritoneal surface with the aim of eliminating residual microscopic peritoneal disease more efficiently than systemic treatment [68]. While a seminal study reported improved survival by adding cisplatin–paclitaxel-based intraperitoneal chemotherapy compared to intravenous only chemotherapy, it nevertheless exhibited severe morbidity [69]. The same trend was reported in subsequent trials [70,71,72]. This increased toxicity led to the limited development of intraperitoneal delivery in aEOC. Nevertheless, the Dutch phase III, open label, multicentric, OVHIPEC (NCT00426257) RCT led to renewed interest in HIPEC [73]. Published in 2018, this trial enrolled 245 patients (ratio 1:1) and evaluated the impact of associating HIPEC with ICS (following 3 cycles of carboplatin plus paclitaxel NACT) in patients with FIGO III aEOC ineligible for PCS or with incomplete surgery. In this trial, the HIPEC procedure consisted of cisplatin (100 mg/m^2^, delivered during a per-surgery 90 min procedure) plus sodium thiosulfate (a nephroprotective agent). Randomization was performed at the time of surgery in cases in which surgery was deemed feasible. Upon ICS +/− HIPEC, three additional carboplatin plus paclitaxel cycles were administered. Survival benefit was significant with HIPEC in both recurrence-free survival (RFS; 14.2 versus 10.7 months; *p* = 0.003) and OS (45.7 vs. 33.9 months; *p* = 0.02). Nevertheless, this study raised several concerns regarding its limitations: exclusion of stage IV diseases, slow recruitment of patients, randomization process occurring after 3 cycles of NACT, lack of stratification for important prognostic factors such as BRCA1/2 status, and initial response rate to NACT [74,75]. In 2022, the Spanish CARCINOHIPEC (NCT02328716) monocentric phase III RCT included 71 patients (ratio 1:1) with FIGO IIIb/c aEOC and reported a non-statistically significant benefit with HIPEC associated with ICS, with a RFS of 18 versus 12 months (*p* = 0.12) [76]. The same year, a bicentric Korean (NCT01091636) RCT included 184 patients (ratio 1:1) and reported negative results regarding RFS [77]. Nevertheless, this study had a different design, as it included both FIGO III and IV stages; furthermore, it included both PCS and ICS surgeries with HIPEC (with PCS representing 41% of cases). As such, mixing two distinct strategies (PCS plus HIPEC and ICS plus HIPEC) probably led to hiding the positive effect of HIPEC on microscopic residues. Furthermore, regarding the locoregional recruitment of these studies, we cannot totally rule out a differential effect of HIPEC depending on ethnicity. The OVHIPEC-2 (NCT03772028) is currently recruiting patients with the aim of specifically assessing PCS and HIPEC in the context of FIGO III stages aEOC [78].

When considering the choice between PCS and ICS plus HIPEC in the context of a patient eligible for primary surgery, current literature is insufficient to favor ICS plus HIPEC. Noteworthy, following the publication of the seminal OVHIPEC study, international guidelines have given contrasted positions: while NCCN considers HIPEC with ICS as a possible strategy for FIGO III aEOC, the ESMO/ESGO consensus stated that HIPEC should not be considered standard therapy and that it should be limited to well-designed prospective RCTs [10,11].

### 3.4. A Patient’s Perspective

Despite its initial underestimation in RCTs and in clinical practice, the patient perspective is of great importance, especially when choosing between NACT and PCS strategies. As a way to objectively determine the patient’s perspective, EORCT QoL questionnaires such as the EORCT QoL QLQ-C30 and QLQ-OV28 are of particular interest [79,80]. Regarding the EORCT 55971 trial, no differences in QoL with QLQ-C30 were reported between PCS and ICS; strikingly, it was also reported that institutions with good QoL compliance also exhibited optimal debulking surgeries in PFS and OS [80]. Following the same trend, no differences regarding QoL with QLQ-C30 were observed in the CHORUS trial [25]. In contrast, in the SCORPION study, mean QLQ-C30 scores in six domains (i.e., emotional functioning, cognitive functioning, nausea/vomiting, dyspnea, insomnia, and hair loss) were statistically and clinically better in the NACT group, although scores in both groups improved over time [81]. Recent meta-analyses (5 RCTs, 1774 patients) exhibited that NACT probably reduces the risk of serious adverse events (notably in the postoperative context), postoperative mortality, and the need for stoma formation [42]. Interestingly, in the phase II, observational, international SOCQER-2 study, patients who received extensive surgical procedures (assessed with a surgical complexity score) only experienced small to moderate decreases in QoL at 6 weeks post-surgery, which resolved by 6–12 months [82].

### 3.5. Molecular Features

Apart from distinguishing diseases with FIGO staging, there is increasing evidence of the molecular features of aEOC and their strong influence on successful surgical procedures. Indeed, although biological parameters may appear unconnected with the outcome of technical procedures such as cytoreductive surgery, recent literature has supported interesting data. While *BRCA2* (and, to a lesser extent, *BRCA1*) mutational status has been associated with a favorable prognosis, its impact on the CC0 rate has been partly elusive so far [83]. An Italian multicenter study on 273 aEOC—of whom 39.2% were germline *BRCA1* or *BRCA2* mutated (*gBRCA1/2**)—reported higher incidences of peritoneal spread without ovarian mass (25.2% vs. 13.9%; *p* = 0.018), of bulky lymph nodes (30.8% vs. 17.5%; *p* = 0.010), and of peritoneal tumor load (laparoscopic predictive index value ≥8; 42.1% vs. 27.1%; *p* = 0.016) in *gBRCA1/2** patients. Of note, no differences between PCS and ICS strategies regarding median PFS were observed in patients with *gBRCA1/2**, in contrast with patients with the wild-type *gBRCA1/2* genotype. More recently, an international retrospective study evaluated the impact of tumoral *BRCA1/2* status (i.e., including both germline and somatic contexts) on clinical presentation and outcome; interestingly, it reported that tumors with *BRCA1/2* mutations exhibited more frequent extrapelvic presentation upon diagnosis and higher optimal debulking rates [84]. Furthermore, based on multi-omics analyses, specific patterns of PCS good and poor responders to NACT have been characterized [85]. As such, apart from a restricted view of enhanced chemosensitivity to platinum salts in the context of *BRCA1/2* mutations, future studies should also consider molecular features from a more global perspective, including medicosurgical optimal strategies.

### 3.6. Complementary Approaches

The past decade has shown the tremendous impact of novel molecules such as bevacizumab and PARPi on aEOC prognosis. Owing to their effect on the adjuvant “maintenance” strategy, it has been hypothesized that they could be used during the neoadjuvant sequence.

Two phase II clinical trials have evaluated the impact of adding bevacizumab to NACT. In the non-comparative randomized ANTHALYA study, 37 patients received NACT and 62 received NACT + bevacizumab for 4 cycles before ICS, followed in all cases by ACT + bevacizumab. A CC0/CC1 rate was observed in 58.6% vs. 51.4% (NACT + bevacizumab and NACT alone, respectively) without added toxicity [86]. In the GEICO 1205 study, 68 patients were randomized between NACT + bevacizumab and NACT [87]. Regarding efficacy, 89% (NACT + bevacizumab) versus 67% (NACT alone) underwent ICS; nevertheless, the CC0/CC1 rate and PFS were similar in both arms.

More recently, PARPi have been evaluated in the neoadjuvant setting. In the phase I NOW trial (NCT03943173), presented at the 2023 Society of Gynecologic Oncology Annual Meeting on Women’s Cancer (SGO-AM), 15 patients with *gBRCA1/2**-bearing unresectable aEOC received 2 cycles (q4w) of neoadjuvant olaparib. Among the 14 patients who underwent surgical debulking, 12 (85.7%) achieved optimal results with no gross residual disease. Ongoing phase II studies, such as the NANT (NCT04507841) trial, which assesses the impact of neoadjuvant niraparib, or the NUVOLA (NCT04261465) trail, which evaluates NACT + olaparib for initially unresectable aEOC, may help understand the relevance of PARPi in the neoadjuvant setting.

In addition to targeted therapies, immune checkpoint inhibitors (ICPI) have made great strides in the last decade for many malignancies. Until recently, trials evaluating ICPI +/− chemotherapy as first-line treatment, such as the phase III IMagyn050/GOG 3015/ENGOT-OV39 trial (NCT03038100) and the JAVELIN Ovarian 100 trial (NCT02718417), failed to improve PFS, despite evidence of synergy between conventional chemotherapy and ICPI [88,89,90]. Nevertheless, preliminary results from more recent studies have given positive signals regarding the benefit of adjuncting ICPI to NACT. For instance, the AdORN trial (NCT03394885), which was presented at the 2021 SGO-AM, evaluated atezolizumab (an anti-PD-L1) in combination with NACT followed by ICS in newly diagnosed aEOC, enrolling 18 patients and reporting optimal cytoreduction in 86% of patients (53% of CC0 and 33% of CC1) [91]. During the 2023 edition of the SGO-AM, results from the phase II KGOG3046/TRU-D (NCT03899610) trial (*n* = 23) were presented. This trial evaluated the combination of NACT with a double-ICPI blockade—using durvalumab (an anti-PD-L1) and tremelimumab (an anti-CTLA4)—for three cycles in the context of initially non-operable FIGO IIIc/IV aEOC. After ICS, three cycles of ACT plus durvalumab were given, followed by durvalumab maintenance for 12 cycles, and the primary endpoint was the 12-month PFS rate. After a median follow-up of 29.2 months, a 12-month PFS rate of 63.6% and a median PFS of 17.5 months were reported. Publications from these seminal trials and upcoming RCTs are highly anticipated.

## 4. Conclusions

Surgical cytoreduction is one of the cornerstones of the initial treatment of aEOC. Both PCS and ICS have been shown to be effective, although there is no clear optimal strategy. Apart from a simple anatomic consideration, effective maximal resection of EOC and optimal medical-surgical strategy depend on several parameters: the caregiver (i.e., surgical skills, experience of the medical-surgical team, and central facilities); the disease itself (i.e., FIGO staging and molecular characteristics); and the patient (i.e., his general status, comorbidities, and preferences). Current strategies that address these three distinct aspects could enable improved personalized care in the context of aEOC.

## Figures and Tables

**Table 2 cancers-15-03556-t002:** Results of the four phase III trials that compared PDS vs. NACT.

Trial	EORTC [24]		CHORUS [25]		JCOG [26]		SCORPION [27]	
Type	Non-inferiority		Non-inferiority		Non-inferiority		Superiority	
Primary objective	OS		OS		OS		Morbidity and PFS ^1^	
Treatment arm (n)	PDS (*n* = 336)	NACT (*n* = 334)	PDS (*n* = 276)	NACT (*n* = 274)	PDS (*n* = 149)	NACT (*n* = 152)	PDS (*n* = 84)	NACT (*n* = 87)
Mean age (y)	62		65		60		56.1	
FIGO Stages III/IV (%)	76.5 ^2^/22.9	75.7/24.3	75/25	75/25	67.1/32.9	69.1/30.9	84.5 ^2^/15.5	90.8/9.2
CT cycles (n)	≥6	3 + 3	6	3 + 3	8	4 + 4	6	6 (3–4 + 2–3)
Patients operated (%)	305 (91%)	292 (87%)	251 (91%)	217 (79%)	147 (98%)	130 (85%)	84 (100%)	74 (85%)
Tumor residue absent (%)	62 (20%)	152 (52%)	39 (17%)	79 (39%)	17 (12%)	83 (64%)	40 (48%)	57 (77%)
Tumor residue < 1 cm (%)	74 (24%)	87 (30%)	57 (25%)	68 (34%)	38 (26%)	24 (18%)	38 (45%)	16 (22%)
Tumor residue ≥1 cm (%)	169 (56%)	53 (18%)	137 (59%)	54 (27%)	92 (62%)	23 (18%)	6 (7%)	1 (1%)
Mean operating time (min)	165	180	120	120	341	273	460.6	253.2
Postoperative mortality ^3^ (%)	8 (2.5%)	2 (0.7%)	14 (5.6%)	1 (0.5%)	1 (0.7%)	0 (0%)	3 (1.7%)	0 (0%)
Postoperative toxicities ^4^ (%)	NA	NA	60 (24%)	30 (14%)	23 (15.6%)	7 (4.6%)	1–6 m: 39 (46.4%) >6 m: 10 (11.9%)	1–6 m: 7 (9.5%) >6 m: 1 (1.4%)
PFS (months)	12	12	10.7	12.0	15.1	16.4	15	14
OS (months)	29	30	22.6	24.1	49.0	44.3	41	43
HR (with CI)	0.98 (90% CI 0.84–1.13)		0.87 (95% CI 0.72–1.05)		1.05 (90.8% CI 0.835–1.326)		1.05 (95% CI 0.77–1.44)	
*p* value	0.01		NA		0.24		0.73	
Non inferiority margin	1.25		1.18		1.161		-	

^1^ PFS was added secondarily (number of subjects necessary was recalculated); ^2^ FIGO IIIc stages only; ^3^ mortality within 28 days after surgery (except for SCORPION study: 30 days); ^4^ grade 3–4 toxicities, evaluated with Common Terminology Criteria for Adverse Events classification. Abbreviations: CI, confidence interval; HR, hazard ratio; NA, not available; NACT, neoadjuvant chemotherapy; OS, overall survival; PFS, progression-free survival.

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
