# Peer review of "Optimal First-Line Medico-Surgical Strategy in Ovarian Cancers: Are We There Yet?"

_cancers, 2023, doi:10.3390/cancers15143556_

Round 1

Reviewer 1 Report

This article focuses on an important topic related to the clinical implications of medico-surgical strategy in ovarian cancers. The title of this article should be modified by identifying the medico-surgical strategy in managing advanced stages of ovarian cancer. The strong point of this study is the highlighting of the personalized therapeutic scheme suitable for the patient, the medical team, and the specialized center. The authors provided adequate details on methodology, evaluation, findings, and investigations. Given the bibliography, it is clear that the authors made a complete review of the literature beforehand. Overall, this manuscript is well-written and documented. 

However, some suggestions could improve the quality of the article:

● It is important to specify the design of this study, the inclusion criteria, as well as the management of metastatic disease.

● What is the prognosis after intraperitoneal hyperthermic chemotherapy and primary or interval cytoreductive surgery in ovarian cancer?

● No information is specified about the preoperative staging of the patients and how the imaging evaluation can influence the therapeutic strategy of the cases.

● The authors could present the information regarding the therapeutic strategy in a small summary table or diagram that will be more attractive to the readers.

● Line 67 completes the unit measure at >2.5.

Kind regards

There are some technical editing mistakes that will have to be corrected.

Author Response

The authors sincerely thank the reviewer for interesting comments. Please find attached responses to all the comments

This article focuses on an important topic related to the clinical implications of medico-surgical strategy in ovarian cancers. The title of this article should be modified by identifying the medico-surgical strategy in managing advanced stages of ovarian cancer. The strong point of this study is the highlighting of the personalized therapeutic scheme suitable for the patient, the medical team, and the specialized center. The authors provided adequate details on methodology, evaluation, findings, and investigations. Given the bibliography, it is clear that the authors made a complete review of the literature beforehand. Overall, this manuscript is well-written and documented. 

However, some suggestions could improve the quality of the article:

  • It is important to specify the design of this study, the inclusion criteria, as well as the management of metastatic disease.

Thanks for this remark, we actually added a table compiling the criteria from RCT’s

  • What is the prognosis after intraperitoneal hyperthermic chemotherapy and primary or interval cytoreductive surgery in ovarian cancer?

Thanks for this remark. Nevertheless, while it is a topical issue, HIPEC is out of scope of our review and would need a dedicated manuscript. We nevertheless included a section in the perspectives.

  • No information is specified about the preoperative staging of the patients and how the imaging evaluation can influence the therapeutic strategy of the cases.

Thanks for this remark, we actually added within the introduction a paragraph stating the limitation of simple FIGO/imaging, positioning initial laparoscopy as the cornerstone of carcinomatosis evaluation

  • The authors could present the information regarding the therapeutic strategy in a small summary table or diagram that will be more attractive to the readers.

Thanks for this remark, we actually added a graphical abstract for giving global information and parameters to consider influencing the surgical strategy

  • Line 67 completes the unit measure at >2.5.

Thanks for this remark, we actually added a table compiling the criteria from RCT’s

Reviewer 2 Report

This is a concise and comprehensive review article regarding primary cytoreductive surgery versus interval debulking surgery following neoadjuvant chemotherapy in the treatment of advanced epithelial ovarian cancer. The issue was discussed from the aspects of randomized controlled trials, real world data, ongoing trials, molecular features, patient’s prospective and complementary approaches.

However, a lot of errors were noted regarding the references. For example, in the context,  there were no references [37] to [56] between the reference [36] in line 158 and [57] in line 224;  the reference [50] “Ovarian Cancer, Version 2.2020, NCCN Clinical Practice Guidelines in Oncology. “ should be updated to version 2023; references were inconsistent with the indicated context in lines 149 (reference [19]), 156 (reference [35]), 158 (reference [36]), 227 (references [64, 65]), 235 (reference [68]), 238 (reference [69]), 239 (reference [70]), 310 (reference [70]), and 321 (reference [42]), etc.

There were also some minor errors. For example, in lines 253 and 254, “with the aim to provide independent and high-quality RWD real- world data Patient data collected from electronic health records. “; in line 309, “The CA-125 ELIMi- nation Rate ConstantK (KELIM)”; in line 316,“ [95CI 0.68-0.84]) “, etc.

In addition, regarding the decision of choosing primary cytoreductive surgery or interval debulking surgery following neoadjuvant chemotherapy, the role of diagnostic laparoscopy should be discussed. Furthermore, the impact of hyperthermic intraperitoneal chemotherapy during interval debulking surgery should also be mentioned.

Author Response

The authors sincerely thank the reviewer for interesting comments. Please find attached responses to all the comments

his is a concise and comprehensive review article regarding primary cytoreductive surgery versus interval debulking surgery following neoadjuvant chemotherapy in the treatment of advanced epithelial ovarian cancer. The issue was discussed from the aspects of randomized controlled trials, real world data, ongoing trials, molecular features, patient’s prospective and complementary approaches.

However, a lot of errors were noted regarding the references. For example, in the context,  there were no references [37] to [56] between the reference [36] in line 158 and [57] in line 224;  the reference [50] “Ovarian Cancer, Version 2.2020, NCCN Clinical Practice Guidelines in Oncology. “ should be updated to version 2023; references were inconsistent with the indicated context in lines 149 (reference [19]), 156 (reference [35]), 158 (reference [36]), 227 (references [64, 65]), 235 (reference [68]), 238 (reference [69]), 239 (reference [70]), 310 (reference [70]), and 321 (reference [42]), etc.

We actually had a problem with zotero leading to misleading numbering of references. We cleared this issue by reinitializing the references.

There were also some minor errors. For example, in lines 253 and 254, “with the aim to provide independent and high-quality RWD real- world data Patient data collected from electronic health records. “; in line 309,

We thank the author for noticing this misprint, we modified accordingly.

 “The CA-125 ELIMi- nation Rate ConstantK (KELIM)”; in line 316,“ [95CI 0.68-0.84]) “, etc.

We thank the author for this remark. Actually, it is the definition of KELIM

In addition, regarding the decision of choosing primary cytoreductive surgery or interval debulking surgery following neoadjuvant chemotherapy, the role of diagnostic laparoscopy should be discussed. Furthermore, the impact of hyperthermic intraperitoneal chemotherapy during interval debulking surgery should also be mentioned. 

We thank the author for these remarks:

Regarding laparoscopy, we emphasized the fact that it is of prime importance for evaluation of resection. Regarding HIPEC, this interesting technique is out of scope of the review, this is why we added it in the third section regarding perspectives.

Reviewer 3 Report

Authors have discussed the pros and cons for PDS vs. IDS surgery in ovarian cancer. The topic is still „hot” and worth of presentation. However, authors should make some improvements to the manuscript .

1/ The 5-class Makar classification and Fagotti score should be presented in the table

2/ The meta-analyses devoted to the same topic should be cited and discussed:

Minerva Med. 2019 Aug;110(4):330-340. doi: 10.23736/S0026-4806.19.06078-6. Primary debulking surgery vs. interval debulking surgery for advanced ovarian cancer: review of the literature and meta-analysis

Benito Chiofalo et al.

Gynecol Oncol 2020 Mar;31(2):e12. doi: 10.3802/jgo.2020.31.e12. Efficacy and safety of neoadjuvant chemotherapy versus primary debulking surgery in patients with ovarian cancer: a meta-analysis

Xiaofeng Lv et al. 

3/ Several papers should be cited and discussed in the paper:

Radiol Oncol. 2018 Sep 11;52(3):307-319. doi: 10.2478/raon-2018-0030.

Primary debulking surgery versus primary neoadjuvant chemotherapy for high grade advanced stage ovarian cancer: comparison of survivals. Borut Kobal et al. 

J Ovarian Res 2021 Mar 27;14(1):49. doi: 10.1186/s13048-021-00801-4.

Choosing the right timing for interval debulking surgery and perioperative chemotherapy may improve the prognosis of advanced epithelial ovarian cancer: a retrospective study. Dengfeng Wang et al. 

Am J Obstet Gynecol. 2016 Apr;214(4):503.e1-503.e6. doi: 10.1016/j.ajog.2015.10.922. Minimally invasive interval debulking surgery in ovarian neoplasm (MISSION trial-NCT02324595): a feasibility study

Salvatore Gueli Alletti et al.

Gynecol Cancer 2017 Jan;27(1):28-36. doi: 10.1097/IGC.0000000000000843.

Patterns of Recurrence and Clinical Outcome of Patients With Stage IIIC to Stage IV Epithelial Ovarian Cancer in Complete Response After Primary Debulking Surgery Plus Chemotherapy or Neoadjuvant Chemotherapy Followed by Interval Debulking Surgery: An Italian Multicenter Retrospective Study. Angiolo Gadducci et al. 

Gynecol Oncol. 2020 Sep;158(3):608-613. doi: 10.1016/j.ygyno.2020.05.041. A multimodality triage algorithm to improve cytoreductive outcomes in patients undergoing primary debulking surgery for advanced ovarian cancer: A Memorial Sloan Kettering Cancer Center team ovary initiative

Alli M Straubhar et al.

Medicine (Baltimore) 2016 Sep;95(36):e4797. doi: 10.1097/MD.0000000000004797.

Effect of neoadjuvant chemotherapy on platinum resistance in stage IIIC and IV epithelial ovarian cancer. Yanlin Luo et al. 

Author Response

The authors sincerely thank the reviewer for interesting comments. Please find attached responses to all the comments

Authors have discussed the pros and cons for PDS vs. IDS surgery in ovarian cancer. The topic is still „hot” and worth of presentation. However, authors should make some improvements to the manuscript .

1/ The 5-class Makar classification and Fagotti score should be presented in the table

We added a Table 1 with Makar classification, Fagotti score and residual CC.

2/ The meta-analyses devoted to the same topic should be cited and discussed:

Minerva Med. 2019 Aug;110(4):330-340. doi: 10.23736/S0026-4806.19.06078-6. Primary debulking surgery vs. interval debulking surgery for advanced ovarian cancer: review of the literature and meta-analysis

Benito Chiofalo et al.

Gynecol Oncol 2020 Mar;31(2):e12. doi: 10.3802/jgo.2020.31.e12. Efficacy and safety of neoadjuvant chemotherapy versus primary debulking surgery in patients with ovarian cancer: a meta-analysis

Xiaofeng Lv et al. 

Both reviews have been added as references and discussed

3/ Several papers should be cited and discussed in the paper:

Radiol Oncol. 2018 Sep 11;52(3):307-319. doi: 10.2478/raon-2018-0030.

Primary debulking surgery versus primary neoadjuvant chemotherapy for high grade advanced stage ovarian cancer: comparison of survivals. Borut Kobal et al. 

J Ovarian Res 2021 Mar 27;14(1):49. doi: 10.1186/s13048-021-00801-4.

Choosing the right timing for interval debulking surgery and perioperative chemotherapy may improve the prognosis of advanced epithelial ovarian cancer: a retrospective study. Dengfeng Wang et al. 

Am J Obstet Gynecol. 2016 Apr;214(4):503.e1-503.e6. doi: 10.1016/j.ajog.2015.10.922. Minimally invasive interval debulking surgery in ovarian neoplasm (MISSION trial-NCT02324595): a feasibility study

Salvatore Gueli Alletti et al.

Gynecol Cancer 2017 Jan;27(1):28-36. doi: 10.1097/IGC.0000000000000843.

Patterns of Recurrence and Clinical Outcome of Patients With Stage IIIC to Stage IV Epithelial Ovarian Cancer in Complete Response After Primary Debulking Surgery Plus Chemotherapy or Neoadjuvant Chemotherapy Followed by Interval Debulking Surgery: An Italian Multicenter Retrospective Study. Angiolo Gadducci et al. 

Gynecol Oncol. 2020 Sep;158(3):608-613. doi: 10.1016/j.ygyno.2020.05.041. A multimodality triage algorithm to improve cytoreductive outcomes in patients undergoing primary debulking surgery for advanced ovarian cancer: A Memorial Sloan Kettering Cancer Center team ovary initiative

Alli M Straubhar et al.

Medicine (Baltimore) 2016 Sep;95(36):e4797. doi: 10.1097/MD.0000000000004797.

Effect of neoadjuvant chemotherapy on platinum resistance in stage IIIC and IV epithelial ovarian cancer. Yanlin Luo et al. 

The authors sincerely thank the reviewer for relevant bibliography. All these references have been added and discussed within our manuscript

Round 2

Author Response

The authors sincerely thank the reviewer for its implication in the reviewing process. Please find attached our responses

In the current review article, the four randomized trials ( CHORUS, JCOG, SCORPION) showed similar survival outcome by either primary cytoreductive surgery or interval debulking in ovarian cancers. However, a review article talking about primary cyto reductive surgery vs. interval debulking in ovarian cancers should not omit the impact role of HIPEC. The OVHIPEC randomized trial (N Engl J Med 2018;378:230 240) demonstrated that “Among patients with stage III epithelial ovarian cancer, the addition of HIPEC to interval cytoreductive surgery resulted in longer recurrence free survival and overall survival than surgery alone and did not result in higher rates of side effects.”. That is to say, adding HIPEC to interval debulking might break the previous concepts that primary cytoreductive surgery and interval debulking resulted in similar survival outcome.

We sincerely apologize, we added the following paragraph

3.3. Hyperthermic Intraperitoneal Chemotherapy

3.3. Hyperthermic Intraperitoneal Chemotherapy

Owing to the fact that disease recurrence primarily involves peritoneal cavity, hyperthermic intraperitoneal chemotherapy (HIPEC) complementary to cytoreductive surgery have been developed. HIPEC relies on enhanced delivery of chemotherapy at the peritoneal surface, with the aim to eliminate residual microscopic peritoneal disease more efficiently than systemic treatment [68]. While a seminal study reported improved survival by adding cisplatin–paclitaxel-based intraperitoneal chemotherapy compared to intravenous only chemotherapy, it nevertheless exhibited severe morbidity [69]. The same trend was reported in subsequent trials [70–72]. This increased toxicity led to limited development of intraperitoneal delivery in aEOC. Nevertheless, the Dutch phase III, open label, multicentric, OVHIPEC (NCT00426257) RCT led to renew interest in HIPEC [73]. Published in 2018, this trial enrolled 245 patients (ratio 1:1) evaluated the impact of associating HIPEC with ICS (following 3 cycles of carboplatin plus paclitaxel NACT) in patients with FIGO III aEOC ineligible to PCS or with incomplete surgery. In this trial, the HIPEC procedure consisted of cisplatin (100mg/m2, delivered during a per-surgery 90 minutes procedure) plus sodium thiosulfate (nephroprotective agent). Randomization was performed at the time of surgery in cases in which surgery deemed to be feasible. Upon ICS +/- HIPEC, three additional carboplatin plus paclitaxel cycles were administered. Survival benefit was significant with HIPEC in both recurrence-free survival (RFS; 14.2 versus 10.7 months; p=0.003) and OS (45.7 vs 33.9 months; p=0.02). Nevertheless, this study raised several concerns regarding its limitations: exclusion of stage IV diseases, slow recruitment of patients, randomization process occurring after 3 cycles of NACT, lack of stratification for important prognostic factors such as BRCA1/2 status and initial response rate to NACT [74,75]. In 2022, the Spanish CARCINOHIPEC (NCT02328716) monocentric phase III RCT included 71 patients (ratio 1:1) with FIGO IIIb/c aEOC and reported a non-statistically significant benefit with HIPEC associated with ICS, with a RFS of 18 versus 12 months (p=0.12) [76]. The same year, a bicentric Korean (NCT01091636) RCT included 184 patients (ratio 1:1) and reported negative results regarding RFS [77]. Nevertheless, this study had a different design, as it included both FIGO III and IV stages; furthermore, it included both PCS and ICS surgeries with HIPEC (with PCS representing 41% of cases). As such, mixing two distinct strategies (PCS plus HIPEC and ICS plus HIPEC) probably led to hide the positive effect of HIPEC on microscopic residues. Furthermore, regarding the locoregional recruitment of these studies, we cannot totally rule out a differential effect of HIPEC depending on ethnicity. The OVHIPEC-2 (NCT03772028) is currently recruiting patients, with the aim to assess specifically PCS and HIPEC in the context of FIGO III stages aEOC [78].  

When considering the choice between PCS and ICS plus HIPEC in the context of a patient eligible to primary surgery, current literature is insufficient to favor ICS plus HIPEC. Noteworthy, following the publication of the seminal OVHIPEC study, international guidelines have given contrasted positions: while NCCN considers HIPEC with ICS as a possible strategy for FIGO III aEOC, ESMO/ESGO consensus stated that HIPEC should not be considered as standard therapy and that it should be limited to well-designed prospective RCT’s [10,11].

There were still errors in the references. For example, reference [ was a retrospective study, not

the indicated SCORPION randomized trial; in reference [ the referred NCCN guideline was of

version 2.2020, but the current updated NCCN guideline was version 2.2023.

We sincerely apologize, we checked again the references one by one. If necessary, please use the pdf provided, as word file uploading may puzzle referencing.

Reviewer 3 Report

The manuscript has been improved significantly, is more informative and easy readable. 

Author Response

The authors sincerely thank the reviewing for his implication in highlighting weaknesses in the initial manuscript, allowing us to improve the global quality of the present review. Best regards